# Abductor Muscle Force after Straight-Stem Compared to Short-Stem Total Hip Arthroplasty through a Modified Direct Lateral Approach: Functional Assessment of 70 Consecutive Patients of a Randomized Controlled Clinical Trial

**DOI:** 10.3390/jcm10061235

**Published:** 2021-03-16

**Authors:** Michael Fuchs, Marie-Anne Hein, Martin Faschingbauer, Mirco Sgroi, Ralf Bieger, Heiko Reichel, Tobias Freitag

**Affiliations:** Department of Orthopaedic Surgery, University of Ulm, Oberer Eselsberg 45, 89081 Ulm, Baden Württemberg, Germany; Michi.fuchs@icloud.com (M.-A.H.); martin.faschingbauer@rku.de (M.F.); Mirco.sgroi@rku.de (M.S.); ralfbieger@hotmail.com (R.B.); heiko.reichel@rku.de (H.R.); tobias.freitag@rku.de (T.F.)

**Keywords:** short stem, muscle force, abductor muscle function, soft tissue damage, total hip replacement

## Abstract

Because of preservation of proximal femoral bone stock and minimized soft tissue trauma, short-stem implants are becoming increasingly important in total hip arthroplasty (THA). The postulated advantage regarding the functional outcome has not been verified. We hypothesized an increased abductor muscle strength by the use of a short-stem design. Seventy consecutive patients of a randomized clinical trial were included. Of these, 67 patients met the inclusion criteria after 12 months. Thirty-five patients received a standard straight stem and 32 patients a short-stem femoral component. All surgeries were performed by a modified direct lateral approach. Isometric muscle strength of the hip abductors was evaluated preoperatively 3 and 12 months after surgery. Harris hip score (HHS) and Western Ontario and McMaster Universities Osteoarthritis Index (WOMAC) scores were evaluated. After three months, there were no differences between the two groups; the abductor force was comparable to the preoperative initial values. After 12 months, a significant increase in muscle strength for the short stem patient group compared to preoperative baseline values was measured (straight-stem THA, 0.09 Nm/kg ± 0.4, *p* = 0.32; short-stem THA, 0.2 Nm/kg ± 0.3, *p* = 0.004). Comparison of the 12-month postoperative total HHS and WOMAC revealed no significant differences between both groups. A significant increase in hip abductor muscle strength 12 months after short-stem THA compared to conventional-stem THA was observed.

## 1. Introduction

During the preparation of the femoral canal in total hip arthroplasty (THA), the abductors of the hip joint sustain a certain degree of tissue alteration. This may result in muscular damage, potentially leading to abductor weakness and gluteal pain [1,2]. Short-stem hip prostheses designs minimize soft tissue injury and provide a bone-preserving implantation technique [3,4]. Curved stem designs allow a calcar-guided implantation without compromising the greater trochanter region. In contrast, most straight stems affect the trochanteric bone stock, potentially leading to significant muscular damage during the implantation process [5]. With regard to the best individual outcome, full recovery of muscular strength is mandatory for postoperative function and patient satisfaction after total hip arthroplasty. Some studies indicate an increased functional outcome after short-stem THA because of a minimally-invasive surgical approach [6,7,8,9]. To date, there has been no investigation evaluating the potentially increased abductor function by the use of objective muscular measurement techniques comparing short- and conventional straight-stem implantation procedures. However, the use of conventional straight- or short-stem designs is still a matter of considerable debate, in particular, because of missing functional middle- and long-term results. The aim of this study was to investigate the postoperative changes of hip abductor muscle strength after short-stem compared to conventional THA. We hypothesized that the use of a short-stem design leads to an increased abductor muscle strength and thus a higher patient satisfaction. The latter was objectified by the use of patient-reported outcome measures (PROMs) illustrating secondary efficacy parameters.

## 2. Material and Methods

### 2.1. Study Design

As part of a larger randomized controlled trial (NCT03147131), 70 consecutive patients were prospectively included after local ethics committee approval (no. 108/10). Informed consent was obtained from all individuals. Patient inclusion and surgical treatments were performed between May 2011 and February 2012. The patient cohort was prospectively identified in a single hospital. Patients aged ≥ 18 years in whom cementless THA was indicated were screened for inclusion. Patients were blinded and randomized for either a straight-stem (CLS, Zimmer, Warsaw, IN, USA, *n* = 35) or a short-stem THA procedure (Fitmore, Zimmer, Warsaw, IN, USA, *n* = 35). Exclusion criteria were: previous surgery in the same hip, femoral fracture, metabolic bone disease, drugs affecting bone quality, and contralateral THA within the study period. The patients were examined preoperatively, at 12-week and 1-year follow-ups.

### 2.2. Patients and Demographics

Three patients of the short stem group were lost to follow up because of absence at the 12-month examination, resulting in a dropout rate of 4%. Thus, a total of 67 patients including 39 men and 28 women, were evaluated. The median age at the time of joint replacement surgery was 58 years (range: 33–74 years; Table 1). No general, local, or surgical complications were observed within the follow-up period. In 66 cases (94%), primary osteoarthritis (OA) was the underlying reason for total joint replacement of the hip. In 4 patients (6%), secondary OA was decisive for subsequent THA (hip dysplasia Crowe type II in 1 case and avascular necrosis of the femoral head (AVN) in 3 cases).

### 2.3. Surgeries and Implant Characteristics

All surgeries were performed by 4 high-volume surgeons with a minimum of 100 annual primary THAs each by the use of a modified direct lateral approach [10]. The gluteus medius was incised along the fiber course to a maximum length of 3 cm to protect the inferior branch of the superior gluteal nerve. The anterior third of the gluteus medius was detached together with the underlying gluteus minimus ventrally to expose the joint capsule. Lengthening of the incision into the vastus lateralis was strictly avoided.

The trochanter-sparing short-stem is made of a titanium alloy and was tapered in three planes with a trapezoidal cross-section (Figure 1a,b and Figure 2a,b). In the proximal half of the component, bony ingrowth is supported by a plasma-coated surface. To restore individual anatomy, the medial curvature of the stem has 3 variations with decreasing radii. The straight and tapered stems are made of a titanium alloy with a grit-blasted surface. The cross-section is rectangular in the sagittal plane, with an additional triple-tapered design and proximal two-dimensional tapered fins. Individual patient anatomy can be addressed by using one of three different neck angles (125, 135, and 145°). The acetabular socket was an Allofit or Trilogy press-fit cup (Zimmer, Warsaw, IN, USA) with a 32 mm ceramic-on-polyethylene bearing in all hips.

All patients were allowed full weight-bearing from the first postoperative day with a standardized exercise protocol. In order to prevent peak loads within the postoperative course, crutches were used for the initial mobilization process until safe gate patterns could be observed. All patients underwent equal postoperative and rehabilitation protocols.

### 2.4. Patient-Reported Outcome Measures (PROM)

Patients referred to subjective pain and mobility levels as self-reported, patient-related outcome measures. The Harris hip score (HHS) and the Western Ontario and McMaster Universities Osteoarthritis Index (WOMAC) were used for the respective evaluation. Questionnaires were collected one day prior to surgery as well as three and 12 months postoperatively.

### 2.5. Functional Assessment

Isometric muscle strength of the hip abductors was evaluated preoperatively, 3 and 12 months after surgery, with a hand-held dynamometer (Lafayette Manual Muscle Test System, model 01163; Lafayette Instrument Company, Lafayette, IN, USA) on standardized landmarks using a routine clinical protocol, which has been previously established in abductor muscle force measurement [11,12]. The dynamometer was placed at the distal lateral femoral condyle with the patient placed in a lateral position (Figure 3). The pad of the resistance arm was centred over the distal lateral femur, comprising 80% of the length from the greater trochanter to the lateral femoral condyle (LC). Strength measurements were expressed in units of torque (Nm) using the distance between the superior anterior spine and the measurement point. To allow a valid between-subject comparison, recorded forces were scaled to body mass [13]. Maximum isometric abductor muscle strength (AMS) was measured for each THA over 3 s at 10 degrees of abduction and neutral rotation to minimize the possible effect of variation in the range of hip movement. In order to minimize the effect of variation on effort, the average value of three readings was calculated. The dynamometer underwent a calibration process at weekly intervals. Functional assessment was performed 1 day preoperatively and 3 and 12 months postoperatively in our outpatient department by a single author, who was adequately trained in AMS measurements and blinded to the implant choice.

### 2.6. Radiographic Evaluation

All patients underwent radiological follow-up examinations on a routine basis 5 days postoperatively as well as 3 and 12 months after THA. The radiological follow-up included standardized digital anteroposterior (AP) radiographs of the pelvis and a lateral view of the affected hip. All radiographs were examined for signs of loosening, for example, radiolucent line formation, respectively bony ingrowth. Additionally, individual offset values were analyzed. Lateral femoral offset (FO) and total offset (TO) of the hip joint were measured pre- and postoperatively on standardized, calibrated anterior posterior radiographs of the pelvis using a landmark-based medical planning software mediCAD^®^ (Hectec GmbH, Altdorf/Landshut, Germany) by a single observer.

Femoral offset was defined as the distance from the centre of rotation of the femoral head to the anatomical femoral axis, and total offset was defined as the distance between the teardrop along the trans-teardrop line and the central femoral axis [14].

### 2.7. Statistics

The Mann-Whitney-U test was used to calculate any statistical difference between the two groups. Wilcoxon’s signed-rank test was used to compare preoperative and postoperative muscle strength as well as the Harris hip score and the WOMAC osteoarthritis index. Data are presented as means with standard deviations and ranges unless otherwise stated. Two-sided *p*-values < 0.05 were considered significant. Statistical analysis was performed using SPSS version 22 (by IBM., Armonk, NY, USA).

## 3. Results

### 3.1. Patient-Reported Outcome Measures

There was a significant improvement in HHS (CLS: *p* < 0.05, Fitmore: *p* < 0.05) and WOMAC scores (CLS: *p* < 0.05, Fitmore: *p* < 0.05) in both groups at the three-month follow-up with no further significant changes up to 12 months (Figure 4a,b). There were no significant differences between the two groups in terms of the HHS and the WOMAC scores either before or after surgery (Table 2).

### 3.2. Functional Assessment

After three months, an increase of 8% with regard to average abductor muscle force was observed for the short-stem group (straight-stem group, −1%, Table 2). After 12 months, respective functional assessments revealed an increase for both stem types compared to preoperative baseline values (short-stem group, +40%; straight-stem group, +13%). Compared to the preoperative muscle force measurements, neither a significant increase nor decrease could be demonstrated three months postoperatively for both stem designs (short-stem *p* = 0.28; straight-stem *p* = 0.34). Short and straight stem designs showed a significant muscle force increase between 3 months compared to 12 months postoperatively (Fitmore: *p* < 0.05, CLS: *p* < 0.05). There was a substantial increase in abductor muscle force after 12 months compared to the preoperative measurements solely for the short-stem group (*p* < 0.05, Figure 5).

### 3.3. Radiographic Evaluation

Within the follow-up period, there were no signs of loosening or osteolysis on conventional X-ray-evaluations. Bony ingrowth without stem subsidence was observed in all cases. All patients showed sufficient integration of the components without signs of substantial implant migration. Total offset as well as limb length was restored within both groups. A slight medialization of the center of rotation was compensated by an increased femoral offset without significant differences in total offset changes between both groups (*p* = 0.47). Femoral offset was equally reconstructed with both stem types without significant differences (*p* < 0.05, Table 2).

## 4. Discussion

The main advantage of short-stem THA is attributable to a bone-stock-sparing preparation, potentially leading to a muscle-preserving implantation of the femoral component [1,3,11,15]. Though some studies indicate an increased functional outcome of short-stem THA, quantitative evaluations comparing the potential differences in abductor muscle force with conventional straight-stem designs are scarce [4,5,6]. By isometric muscle force measurements of the hip abductors at different timepoints in the setting of a standardized clinical protocol, we were unable to detect significant functional differences between the evaluated groups. A substantial increase in hip abductor muscle force was solely recorded for short-stem THA patients 12 months postoperatively compared to baseline values. With regard to individual joint geometry, femoral offset was sufficiently restored with both stem designs.

McGrory et al. highlighted the importance of adequate offset reconstruction with regard to abductor muscle strength and functional outcomes [16]. Schmidutz et al. compared the biomechanical reconstruction after conventional vs. short-stem THA with an analysis of 50 consecutive surgeries per group [4]. The authors described an average femoral offset increase of 6.2 mm after short-stem THA without assumed functional impairments. Our data also revealed a slightly increased femoral offset of 4.3 mm for short-stem and 3.8 mm for conventional THA. In contrast to the present study, the authors did not analyze total offset reconstruction. Against this background, we could demonstrate that the latter was not substantially impaired within the analyzed patient groups. This is attributable to the fact that a slight medialization of the center of rotation, resulting in decreased acetabular offset was addressed by an increased femoral offset reconstruction without significant impact on total offset values. Given this thought, individual hip biomechanics can be sufficiently restored with both implant designs.

With regard to hip abductor muscle strength after short-stem THA, Kamada et al. conducted a functional analysis of 32 patients after an average follow-up of 46 months [11]. Regardless of the surgical approach (lateral vs. posterolateral), the abductor muscle strength ratio revealed a functional restitution of 91–92% compared to the contralateral non-operated hip. Kamada et al. had to exclude a large number of patients because of hip osteoarthritis in the non-operated joints that served as control groups. Thus, the objective consideration of muscle force measurements might be debatable because of a relatively small number of evaluated cases. Furthermore, Kamada et al. displayed functional results only as a ratio and not as quantitative measurements. The main conclusion states that with reduced femoral offset, a concomitant decrease in abductor muscle strength is evident. This fact is unsurprising as previous studies impressively demonstrated the importance of sufficient offset reconstruction on postoperative muscle function [16,17,18]. In contrast to the present study, there was no comparison of standard vs. short-stem THA with regard to functional outcomes. Given this fact, a conclusion highlighting the potential advantages of short-stem implant designs that might be accompanied with reduced muscular damage during the implantation process is not possible.

Regarding functional as well as radiographic analyses, we aimed to analyze patient-reported outcome measures. No differences could be observed in terms of the Harris hip score or the WOMAC at the respective timepoints. This does not reflect the results of the muscle force measurements. Both the HHS and the WOMAC scores evaluate the patient’s condition of daily life activities. The different increase in maximum abductor muscle strength between the two groups may not be captured by the two scores and will possibly become apparent evaluating higher activity levels. However, pre- and postoperative HHS scores are comparable with those published in the literature, and both scores are among the most frequently found in the literature [6,18,19].

Some limitations of our study must be mentioned. First, the follow-up examinations focused on a one-year interval. As a consequence, the obtained results assess only a limited time period after THA. Second, our study population focused on a small patient cohort. Therefore, individual risk calculations with subgroup analysis to identify potential patient-associated parameters in terms of poorer outcomes were not undertaken. Given the small sample size, it is difficult to relate the functional differences one year post-operation solely to stem design, despite equal postoperative and rehabilitation protocols in both groups. Third, the modified direct lateral surgical approach, which was performed in all patients may be debatable with regard to iatrogenic muscular damage potentially leading to impaired functional results. Thus, the observed differences in muscle strength one year post-operation may also be due to surgical-related reasons. Accordingly, Mueller et al. conducted a study that analyzed the abductor muscle damage dependent on the surgical approach [8]. The authors evaluated functional and MRI-based radiographic results after anterolateral vs. direct lateral THA in 44 patients 3 and 12 months post-surgery. Though abductor muscle damage occurred in both patient groups, they concluded that, by the use of a muscle-sparing anterolateral approach, intraoperative muscular damage can be minimized and concomitant better functional results may be achieved. In contrast to the study by Mueller et al., Greidanus et al. did not find superior functional outcomes after anterolateral THA compared to the direct lateral or posterolateral approach in 135 patients with a minimum follow-up of 24 months [19]. Furthermore, there are studies in the literature indicating that a muscle-sparing approach does not necessarily lead to a higher hip abductor muscle strength one year post-operation [5,20]. The reported studies highlight the fact that there is still an ongoing debate in this context. However, short stems are not necessarily associated with an increased postoperative muscle strength because of many other influencing factors such as surgical related muscular damage or patient body mass index. The majority of studies dealing with hip abductor muscle force measurement after conventional or short-stem THA are conducted via a retrospective study design or lack a control group [5,7,11,15,21,22]. To the best of our knowledge, this is the first study highlighting this issue on the basis of a randomized controlled clinical trial.

## 5. Conclusions

The evaluated short-stem design offers promising results that may contribute to enhanced recovery programs. Conventional THA still can be viewed as a reliable and suitable option in restoring individual hip geometrics and gains excellent results with regard to patient-reported outcome measures.

## Figures and Tables

**Figure 1 jcm-10-01235-f001:**
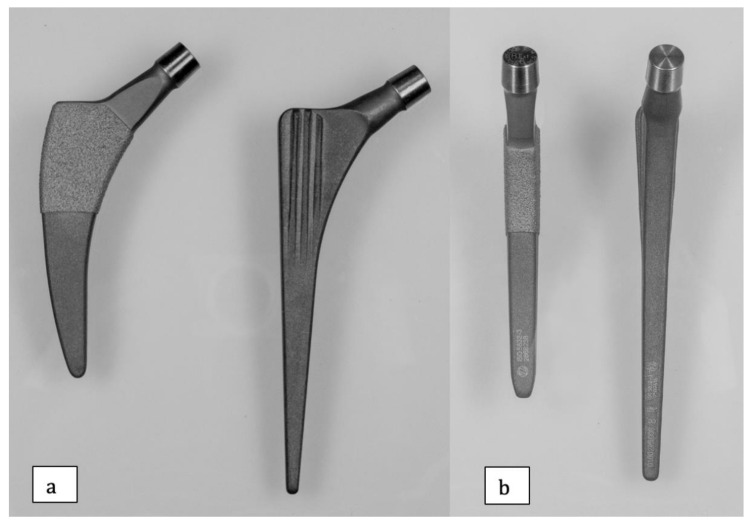
(**a**,**b**) Femoral component designs. Anteroposterior (**a**) and sagittal profiles (**b**) of the analyzed stems. In both figures, the Fitmore stem is shown left and the CLS stem is shown right.

**Figure 2 jcm-10-01235-f002:**
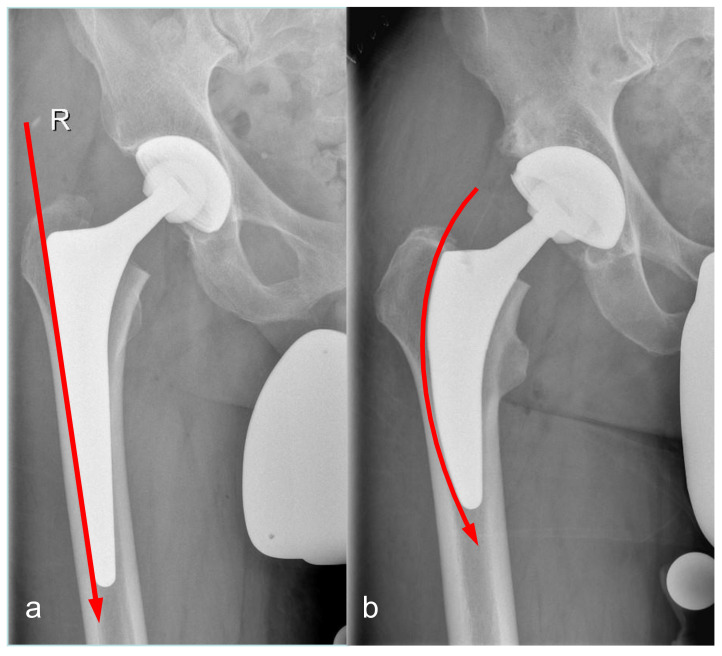
(**a**,**b**) Postoperative X-ray. Anterio-posterior hip X-ray of the CLS (**a**) and Fitmore (**b**) stem. Displayed lines illustrate the trochanter sparing geometry of the Fitmore stem during the implantation process.

**Figure 3 jcm-10-01235-f003:**
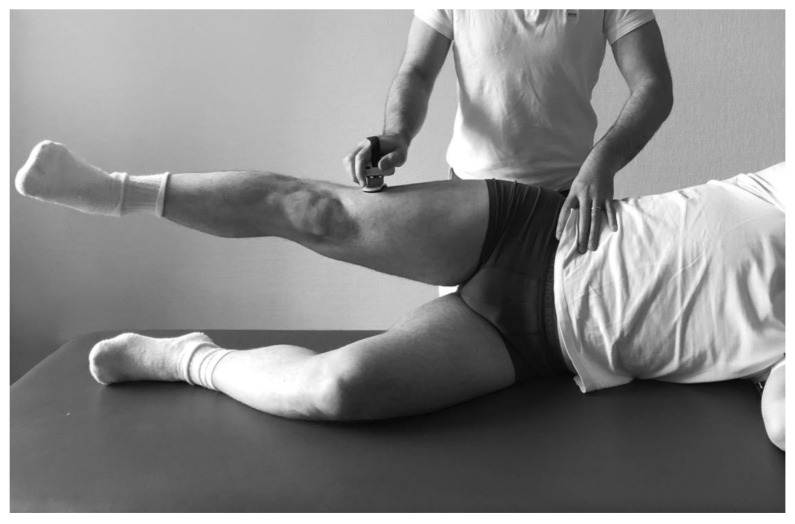
Clinical assessment of isometric hip abduction muscle force in neutral rotation with the use of a handheld dynamometer.

**Figure 4 jcm-10-01235-f004:**
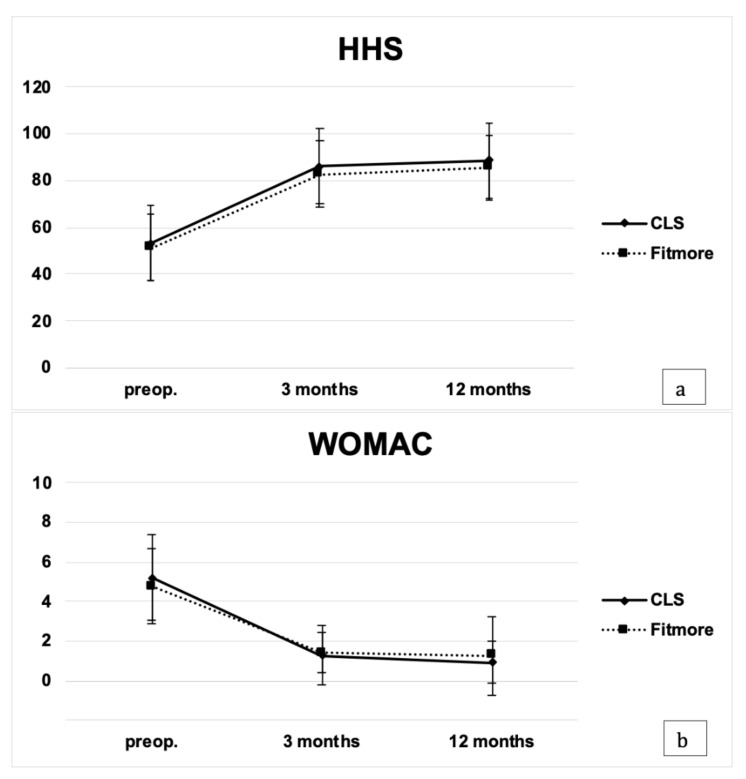
(**a**,**b**) Patient-reported outcome measures. Harris hip scores (**a**) and WOMAC scores (**b**) from baseline up to 12 months follow-up for both stem types (preop.—preoperative).

**Figure 5 jcm-10-01235-f005:**
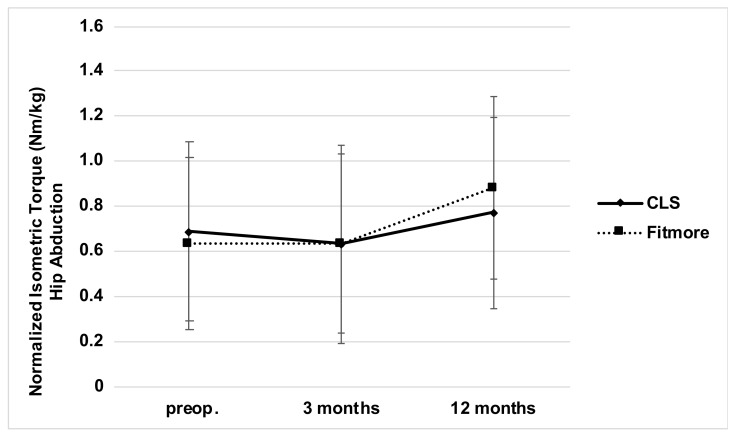
Hip abductor muscle force measurement for both stem types at baseline and postoperative time intervals.

**Table 1 jcm-10-01235-t001:** Patient characteristics.

	CLS (35 hips)	Fitmore (32 hips)	*p*-Value
Age at surgery (years)	59 ± 8 (42–71)	56 ± 10 (33–74)	0.36
Height	170 ± 9 (156–188)	169 ± 7 (153–188)	0.77
Weight (kg)	84 ± 19 (62–138)	82 ± 16 (53–116)	0.94
Body mass index (kg/m^2^)	29 ± 5 (22–45)	29 ± 5 (21–37)	0.89
Gender (F:M)	16:19	15:17	0.39
Affected side (L:R)	14:21	14:18	0.53

Data are presented as mean ± SD; F, female; M, male; L, left, R, right.

**Table 2 jcm-10-01235-t002:** Muscle force measurements, offset values, and patient-reported outcome measures at baseline and postoperative time intervals.

	CLS (35 hips)	Fitmore (32 hips)	*p*-Value
Abductor muscle strength preop. (Nm/kg)	0.68 ± 0.41	0.63 ± 0.38	0.79
Abductor muscle strength at 3 months (Nm/kg)	0.67 ± 0.39	0.68 ± 0.42	0.48
Abductor muscle strength at 12 months (Nm/kg)	0.77 ± 0.43	0.88 ± 0.41	0.38
HHS preop.	53.4 ± 15.0	51.3 ± 14.4	0.50
HHS at 3 months	82.8 ± 14.3	86.1 ± 18.7	0.20
HHS at 12 months	88.8 ± 11.9	85.5 ± 14.8	0.48
WOMAC preop.	5.2 ± 2.2	4.8 ± 1.9	0.38
WOMAC at 3 months	1.3 ± 1.5	1.4 ± 1.0	0.24
WOMAC at 12 months	0.9 ± 1.0	1.3 ± 2.0	0.71
Change femoral offset (mm)	3.8 ± 6.6	4.3 ± 5.5	0.16
Change total offset (mm)	0.3 ± 9.6	0.1 ± 9.5	0.47

Data are presented as means ± SDs; preop.—preoperative; HHS—Harris hip score; WOMAC—Western Ontario and McMaster Universities Osteoarthritis Index.

## Data Availability

The data that support the findings of this study are available from the last author (T.F.), upon reasonable request.

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
