# Peer review of "Abductor Muscle Force after Straight-Stem Compared to Short-Stem Total Hip Arthroplasty through a Modified Direct Lateral Approach: Functional Assessment of 70 Consecutive Patients of a Randomized Controlled Clinical Trial"

_jcm, 2021, doi:10.3390/jcm10061235_

Round 1
Reviewer 1 Report
Dear authors and editors.
A quick search give me:
ANCT03147131: Periprosthetic Bone Mineral Density Changes After Implantation Of A Short Hip Stem Compared To A Straight Stem
"After power analysis, 140 consecutive patients were prospectively included in the randomisation protocol receiving either a Fitmore short or a CLS straight stem. The short stem was assigned in 57 (37% females) cases and the straight stem in 83 (38% females) hips. Periprosthetic bone mineral density was measured before surgery, 7 days, 3, 12 and 60 months postoperatively, using dual energy x-ray absorptiometry (DEXA). "
Publications registered in clinical trial on the NCT:
Freitag T, Hein MA, Wernerus D, Reichel H, Bieger R. Bone remodelling after femoral short stem implantation in total hip arthroplasty: 1-year results from a randomized DEXA study. Arch Orthop Trauma Surg. 2016 Jan;136(1):125-30. doi: 10.1007/s00402-015-2370-z. Epub 2015 Nov 27.Meyer JS, Freitag T, Reichel H, Bieger R. Mid-term gender-specific differences in periprosthetic bone remodelling after implantation of a curved bone-preserving hip stem. Orthop Traumatol Surg Res. 2020 Dec;106(8):1495-1500. doi: 10.1016/j.otsr.2020.04.023. Epub 2020 Oct 31.
This current manuscript is NOT a RCT.
This is an observational study BASED on a RCT - and the manuscript must follow STROBE guidelines. Only 70 of original 140 patients are reported.
When the manuscript has been accurately formatted, I will be happy to review again.
Author Response
Point-by-point response to the comments of Reviewer 1 on the manuscript “Abductor muscle force after straight-stem compared to short-stem total hip arthroplasty: Functional assessment of 70 patients of a randomized controlled clinical trial”
(invited resubmission jcm-1091637)
Comment Reviewer 1:
This current manuscript is NOT a RCT. This is an observational study BASED on a RCT - and the manuscript must follow STROBE guidelines. Only 70 of original 140 patients are reported.
Response: Thank you for this valuable comment. We clearly see the reviewer’s point and appreciate the straight message. Indeed, we evaluated the first consecutive 70 patients of the study NCT03147131. The primary question of the study was to investigate the changes in bone mineral density after implantation of a cementless short compared to a conventional stem. 140 consecutive patients without exclusion criteria were included for this purpose. Secondary outcome measures included PROMs and measurements of abductor muscle strength. When planning the study, we assumed a greater difference in means of the two groups for muscle force than for the DEXA measurements. Therefore, the number of cases for the bone mineral density analyses differ. Taking other RCT`s with similar question into account, the assumed number of cases seems plausible (Jensen C et al. “Recovery in mechanical muscle strength following resurfacing vs standard total hip arthroplasty – a randomized clinical trial”, Osteoarthritis Cartilage 2011, DOI: 10.1016/j.joca.2011.06.011; Krych A J et al. “No benefit of the two-incision THA over mini-posterior THA: a pilot study of strength and gait”, Clin Orthop Relat Res. 2010, DOI: 10.1007/s11999-010-1660-6; Winther S B et al. “A randomized controlled trial on maximal strength training in 60 patients undergoing total hip arthroplasty”, Acta Orthop. 2018, DOI: 10.1080/17453674.2018.1441362.) To point this out more clearly, we added the following amendment within the methods section: (Line 64-65: As part of a larger randomized controlled trial (NCT03147131), 70 patients were prospectively included after local ethics committee approval (No. 108/10).) For reasons of clarity and in order to clearly state that muscle strength was not the primary outcome measure of the study, we specified the title as follows: (Line 2-5: “Abductor muscle force after straight-stem compared to short-stem total hip arthroplasty: Functional assessment of 70 patients of a randomized controlled clinical trial”). However, we believe that study design corresponds to a randomized clinical trial. By addressing the above mentioned issues, we hope to meet the reviewer’s expectations.

Reviewer 2 Report
Other reasons could influence your results.
Author Response
Point-by-point response to the comments of Reviewer 2 on the manuscript “Abductor muscle force after straight-stem compared to short-stem total hip arthroplasty: Functional assessment of 70 patients of a randomized controlled clinical trial”
(invited resubmission jcm-1091637)
Specific comments Reviewer 2:
Other reasons could influence your results.
Response: We completely agree with the reviewer. Please see below (2. Conclusions):
As mentioned within the discussion section, we agree with the reviewer and highlighted the potential other reasons for an increased abductor muscle strength, focusing on the surgical approach or patient specific parameters (Line 238 – 241: “Third, the standard lateral surgical approach, which was performed in all patients might be debatable with regard to iatrogenic muscular damage potentially leading to impaired functional results. Thus, the observed differences in muscle strength 1 year postoperatively might also be due to surgical-related reasons.”; Line 250 – 252: “However, short stems are not necessarily associated with an increased postoperative muscle strength due to many other influencing factors such as surgical related muscular damage or patient body mass index.”)
General comments Reviewer 2:
- Research Design:
Response: Thank you for your assessment. For reasons of clarity, we changed the description of our study design. Against this background, we evaluated the first consecutive 70 patients of the study NCT03147131. Taking other RCT`s with similar question into account, the assumed number of cases seems plausible (Jensen C et al. “Recovery in mechanical muscle strength following resurfacing vs standard total hip arthroplasty – a randomized clinical trial”, Osteoarthritis Cartilage 2011, DOI: 10.1016/j.joca.2011.06.011; Krych A J et al. “No benefit of the two-incision THA over mini-posterior THA: a pilot study of strength and gait”, Clin Orthop Relat Res. 2010, DOI: 10.1007/s11999-010-1660-6; Winther S B et al. “A randomized controlled trial on maximal strength training in 60 patients undergoing total hip arthroplasty”, Acta Orthop. 2018, DOI: 10.1080/17453674.2018.1441362.) To point this out more clearly, we added the following amendment within the methods section: (Line 61-62: As part of a larger randomized controlled trial(NCT03147131), 70 patients were prospectively included after local ethics committee approval (No. 108/10).) For reasons of clarity and in order to clearly state that muscle strength was not the primary outcome measure of the study, we specified the title as follows: (Line 2-5: “Abductor muscle force after straight-stem compared to short-stem total hip arthroplasty: Functional assessment of 70 patients of a randomized controlled clinical trial”). However, we believe that study design corresponds to a randomized clinical trial. By addressing the above mentioned changes, we hope to meet the reviewer’s expectations.
- Conclusions:
Response: Addressing this very valid point of the stated conclusions, we would like to pass on the information that we weakened the message that short-stem THA might increase abductor muscle strength due to a minimized soft tissue trauma (Line 31-32: A significant increase in hip abductor muscle strength 12 months after short-stem THA compared to conventional-stem THA was observed.) As mentioned within the discussion section, we agree with the reviewer and highlighted the potential other reasons for an increased abductor muscle strength, focusing on the surgical approach or patient specific parameters (Line 238 – 241: “Third, the standard lateral surgical approach, which was performed in all patients might be debatable with regard to iatrogenic muscular damage potentially leading to impaired functional results. Thus, the observed differences in muscle strength 1 year postoperatively might also be due to surgical-related reasons.”; Line 250 – 252: “However, short stems are not necessarily associated with an increased postoperative muscle strength due to many other influencing factors such as surgical related muscular damage or patient body mass index.”)

Reviewer 3 Report
I would like to admire the authors' effort for this study. Their clinical question is very interesting and worth being investigated.
Introduction
I agree with the authors that intraoperative muscle injury can influence postoperative muscle strength. However, the authors should consider other factors that influence the muscle strength. Especially, the authors should focus on the stem's bony ingrowth to femur. The stability of femoral stem and where the bone ingrowth happens are very important issue that can influence the abductor muscle strength.
Methods
The authors should add the section for approach. The details of approach are unclear. They evaluated only femoral offset in radiographic analysis. Again, they should add some evaluation about bone ingrowth and stress shielding. They did not mention their sample size calculation. How much did the authors think is significant difference of abductor muscle strength? As this study is RCT, sample size should be defined based on such factors.
Discussion
It is unclear what factors did the authors think influence the difference of abductor muscle strength at po 1 year? The authors seem to advocate this difference was due to less invasion of short stem. But if so, I think the difference can be seen at po 3 months as well.
The authors' motivation to clarify the influence of stem design to postoperative muscle strength is worth being investigated. However, I am afraid this study was conducted in some poor condition (It seems that sample size calculation to find clinically significant difference was not conducted before study began.) They admitted their small sample size. Why did they add more cases despite this is prospective RCT? At least, the authors should avoid to conclude the shorter stem is better for postoperative muscle strength.
Author Response
Point-by-point response to the comments of Reviewer 3 on the manuscript “Abductor muscle force after straight-stem compared to short-stem total hip arthroplasty: Functional assessment of 70 patients of a randomized controlled clinical trial”
(invited resubmission jcm-1091637)
Specific comments Reviewer 3:
- Reviewer 3: I agree with the authors that intraoperative muscle injury can influence postoperative muscle strength. However, the authors should consider other factors that influence the muscle strength. Especially, the authors should focus on the stem's bony ingrowth to femur. The stability of femoral stem and where the bone ingrowth happens are very important issue that can influence the abductor muscle strength.
Response: Thank you for pointing out this very relevant aspect. We agree with the reviewer that the stability of the femoral stem and it’s ingrowth to the femur are relevant parameters in terms of abductor muscle strength. The investigated Fitmore stem is a trochanter-sparing femoral short stem made of a titanium alloy with a plasma- coated surface in the proximal part to support bony ingrowth. The stem is classified as type 4 according to Khanuja et al. with a shortened conventional design with primary fixation in the proximal metaphysis (Khanuja HS, Banerjee S, Jain D, et al. Short bone-conserving stems in cementless hip arthroplasty. J Bone Joint Surg Am 2014; 96: 1742–1752.). Due to the fact that the evaluated short-stem has proven to show similar axial mid-term subsidence compared to conventional straight-stems, we did not further highlight this aspect (Freitag et al.,. Mid-term migration analysis of a femoral short-stem prosthesis: a five-year EBRA-FCA-study. Hip international: 1120700018772277, 2018). This can be justified by the fact that primary stability, osseointegration and subsidence have been investigated in previous studies illustrating that these parameters are comparable to conventional stems (Freitag et al. Migration pattern of a femoral short-stem prosthesis: a 2-year EBRA-FCA-study. Arch Orthop Trauma Surg 134(7): 1003, 2014).
By clarifying this cause for thought, we hope to meet the reviewer’s expectations.
- Reviewer 3: The authors should add the section for approach. The details of approach are unclear. They evaluated only femoral offset in radiographic analysis. Again, they should add some evaluation about bone ingrowth and stress shielding. They did not mention their sample size calculation. How much did the authors think is significant difference of abductor muscle strength? As this study is RCT, sample size should be defined based on such factors.
Response: We clearly see the reviewer’s point and appreciate the straight message. With regard to the surgical technique, a standard lateral transguteal approach was performed (Bauer R, Kerschbaumer F, Poisel S, et al. The transgluteal approach to the hip joint. Arch Orthop Trauma Surg 1979; 95: 47–49.) We added this information within the methods section (Line 80-81: All patients underwent a standardized lateral transgluteal surgical approach.) Additionally, this circumstance was discussed as a limitation (line 237-238). With regard to offset evaluation, lateral femoral offset (FO) as well as total offset (TO) of the hip joint were measured pre- and postoperatively on standardized, calibrated anterior posterior radiographs of the pelvis (Line 130 – 136). The design of the evaluated stem supports bony ingrowth in its proximal half (please see line 73-74). With respect to the very valid point of sample size calculation, it has to be mentioned that we evaluated the first consecutive 70 patients of the study NCT03147131. The primary question of the study was to investigate the changes in bone mineral density after implantation of a cementless short compared to a conventional stem. 140 consecutive patients without exclusion criteria were included for this purpose.
Here, we were able to show that one year after surgery, both stems showed an implant-specific periprosthetic bone remodeling with a trend towards a more pronounced proximal load transfer after short stem implantation than with a straight stem (Freitag et al., . Bone remodelling after femoral short stem implantation in total hip arthroplasty: 1-year results from a randomized DEXA study. Arch Orthop Trauma Surg 136(1): 125, 2016).
Secondary outcome measures included PROMs and measurements of abductor muscle strength. When planning the study, we assumed a greater difference in means of the two groups for muscle force than for the DEXA measurements. Therefore, the number of cases for the bone mineral density analyses differ. Taking other RCT`s with similar question into account, the assumed number of cases seems plausible (Jensen C et al. “Recovery in mechanical muscle strength following resurfacing vs standard total hip arthroplasty – a randomized clinical trial”, Osteoarthritis Cartilage 2011, DOI: 10.1016/j.joca.2011.06.011; Krych A J et al. “No benefit of the two-incision THA over mini-posterior THA: a pilot study of strength and gait”, Clin Orthop Relat Res. 2010, DOI: 10.1007/s11999-010-1660-6; Winther S B et al. “A randomized controlled trial on maximal strength training in 60 patients undergoing total hip arthroplasty”, Acta Orthop. 2018, DOI: 10.1080/17453674.2018.1441362.) To point this out more clearly, we added the following amendment within the methods section: (Line 63-64: As part of a larger randomized controlled trial (NCT03147131), 70 patients were prospectively included after local ethics committee approval (No. 108/10).) For reasons of clarity, we specified the title as follows: (Line 2-5: “Abductor muscle force after straight-stem compared to short-stem total hip arthroplasty: Functional assessment of 70 patients of a randomized controlled clinical trial”). However, we believe that study design corresponds to a randomized clinical trial.
- Reviewer 3: It is unclear what factors did the authors think influence the difference of abductor muscle strength at po 1 year? The authors seem to advocate this difference was due to less invasion of short stem. But if so, I think the difference can be seen at po 3 months as well.
Response: Addressing this very valid point of the stated conclusions and influencing factors, we would like to pass on the information that we attenuated the message that short-stem THA might increase abductor muscle strength due to a minimized soft tissue trauma (Line 31-32: A significant increase in hip abductor muscle strength 12 months after short-stem THA compared to conventional-stem THA was observed.) As mentioned within the discussion section, we agree with the reviewer and highlighted the potential other reasons for an increased abductor muscle strength, focusing on the surgical approach (Line 237 – 240: Third, the standard lateral surgical approach, which was performed in all patients might be debatable with regard to iatrogenic muscular damage potentially leading to impaired functional results. Thus, the observed differences in muscle strength 1 year postoperatively might also be due to surgical-related reasons.). The authors agree with the reviewer, that also differences after 3 months might be conclusive, however, they were not observed.
- Reviewer 3: The authors' motivation to clarify the influence of stem design to postoperative muscle strength is worth being investigated. However, I am afraid this study was conducted in some poor condition (It seems that sample size calculation to find clinically significant difference was not conducted before study began.) They admitted their small sample size. Why did they add more cases despite this is prospective RCT? At least, the authors should avoid to conclude the shorter stem is better for postoperative muscle strength.
Response: As mentioned before, we evaluated the first consecutive 70 patients of the study NCT03147131. Considering other RCT`s with similar questions, the assumed number of cases seems feasible (Jensen C et al. “Recovery in mechanical muscle strength following resurfacing vs standard total hip arthroplasty – a randomized clinical trial”, Osteoarthritis Cartilage 2011, Winther S B et al. “A randomized controlled trial on maximal strength training in 60 patients undergoing total hip arthroplasty”, Acta Orthop. 2018, DOI: 10.1080/17453674.2018.1441362.) We thank the reviewer for the suggestion to weaken the basic message and changed the manuscript accordingly to clarify that shorter stems not necessarily imply an increased postoperative muscle strength
(Line 250-252: “However, short stems are not necessarily associated with an increased postoperative muscle strength due to many other influencing factors such as surgical related muscular damage or patient body mass index.”; Line 258-261: “The evaluated short stem design offers promising results that might contribute to enhanced recovery programs. Conventional THA still can be seen as reliable and suitable option in restoring individual hip geometrics and gains excellent results with regard to patient reported outcome measures.”)

Round 2
Reviewer 1 Report
First of all, I do really believe this manuscript contain information worth of publishing - however, not in its current form.
I personally appreciate the conclusion that new is not better - that the believed design approvements we encounter (often), may not reflect clinical improvements, and that this needs to be evaluated in studies such as this!
However, there are still need for improvement in the manuscript to ensure adequate presentation of this study.
I appreciate the clarifications made by the authors in regard to the concept of this being an RCT.
I would like if the title reflects this even more. As I understand it the 70 patients in this observational study (and it is a such) are drawn consecutively as the first 70 patients included in the RCT - this could be written in the title, and also it should be emphasized that these patients are taken from within an RCT. And also the surgical approach should be included.
The abstract also still needs formatting in this regards.
This also reflects in the method section. I encouraged the authors to adhere to the STROBE guidelines. I do not believe the authors sufficiently describes the flow of patients into the study, still. Please be more rigerous in describing how your patient sample were drawn, and I do not agree with the authors that "However, we believe that study design
corresponds to a randomized clinical trial. " . Either you have a randomised design with an appropriate sample size calculation based on your a priori decided primary outcome parameter or you have an observational study - there is no in-between!
I would also like subheadings to be used in the method section, eg study design, study population, surgical technique etc
Is the median age of 58 normal in this center? The average age in DK is almost 10 years older.
I actually feels that if the authors could restrict the patient population to include patients from 40+ this would make more clinical sense to me in regards to the purpose - probably this will only discard 2-4 patient in total and will only strength the conclusion.
why is there no p-values in the last 2 parameters in table 1?
Also please remove redundant info in table and text.
I need the authors to comment in the discussion on whether the PROMs used actually captures what hip abductor deficiency leads to in elderly patients.
And especially do the authors need to reflect on the surgical approach used and how this could influence the findings. They describe that they use a transgluteal approach and then they perform a study to see if the femur replacement affects gluteal performance (what about the surgical trauma)
Author Response
Point-by-point response to the comments of Reviewer 1 on the manuscript “Abductor muscle force after straight-stem compared to short-stem total hip arthroplasty: Functional assessment of 70 patients of a randomized controlled clinical trial”
(invited resubmission jcm-1091637 – ROUND 2)
Comment Reviewer 1:
First of all, I do really believe this manuscript contain information worth of publishing - however, not in its current form.
I personally appreciate the conclusion that new is not better - that the believed design approvements we encounter (often), may not reflect clinical improvements, and that this needs to be evaluated in studies such as this!
However, there are still need for improvement in the manuscript to ensure adequate presentation of this study.
I appreciate the clarifications made by the authors in regard to the concept of this being an RCT.
I would like if the title reflects this even more. As I understand it the 70 patients in this observational study (and it is a such) are drawn consecutively as the first 70 patients included in the RCT - this could be written in the title, and also it should be emphasized that these patients are taken from within an RCT. And also the surgical approach should be included.
The abstract also still needs formatting in this regards.
This also reflects in the method section. I encouraged the authors to adhere to the STROBE guidelines. I do not believe the authors sufficiently describes the flow of patients into the study, still. Please be more rigerous in describing how your patient sample were drawn, and I do not agree with the authors that "However, we believe that study design
corresponds to a randomized clinical trial. " . Either you have a randomised design with an appropriate sample size calculation based on your a priori decided primary outcome parameter or you have an observational study - there is no in-between!
I would also like subheadings to be used in the method section, eg study design, study population, surgical technique etc
Is the median age of 58 normal in this center? The average age in DK is almost 10 years older.
I actually feels that if the authors could restrict the patient population to include patients from 40+ this would make more clinical sense to me in regards to the purpose - probably this will only discard 2-4 patient in total and will only strength the conclusion.
why is there no p-values in the last 2 parameters in table 1?
Also please remove redundant info in table and text.
I need the authors to comment in the discussion on whether the PROMs used actually captures what hip abductor deficiency leads to in elderly patients.
And especially do the authors need to reflect on the surgical approach used and how this could influence the findings. They describe that they use a transgluteal approach and then they perform a study to see if the femur replacement affects gluteal performance (what about the surgical trauma).
Point-by-point response to the comments of Reviewer 1 on the manuscript “Abductor muscle force after straight-stem compared to short-stem total hip arthroplasty: Functional assessment of 70 patients of a randomized controlled clinical trial”
(invited resubmission jcm-1091637 – ROUND 2)
Specific comments Reviewer 1:
1.) Reviewer 1: I would like if the title reflects this even more. As I understand it the 70 patients in this observational study (and it is a such) are drawn consecutively as the first 70 patients included in the RCT - this could be written in the title, and also it should be emphasized that these patients are taken from within an RCT. And also the surgical approach should be included. The abstract also still needs formatting in this regards.
Response: We clearly see the reviewer’s point and appreciate the straight message. According to the comments we have adapted title and abstract to make the design of the study more recognizable (Line 2-6: Abductor muscle force after straight-stem compared to short-stem total hip arthroplasty through a modified direct lateral approach: Functional assessment of 70 consecutive patients of a randomized controlled clinical trial, Line 21-24: Seventy consecutive patients of a randomized clinical trial were included. Of these, 67 patients met the inclusion criteria after 12 months. Thirty-five patients received a standard straight stem and 32 patients a short stem femoral component. All surgeries were performed by a modified direct lateral approach).
2.) Reviewer 1: This also reflects in the method section. I encouraged the authors to adhere to the STROBE guidelines. I do not believe the authors sufficiently describes the flow of patients into the study, still. Please be more rigerous in describing how your patient sample were drawn. I would also like subheadings to be used in the method section, eg study design, study population, surgical technique etc..
Response: Thank you for your comment. To clarify the patient flow, we have restructured the chapter “methods”. We also provided additional information in accordance with the comments, strobe guidelines were adhered to (Line 68-135: Subchapters “study design”, “patients and demographics”, “surgeries and implant characteristics”).
3.) Reviewer 1: Is the median age of 58 normal in this center? The average age in DK is almost 10 years older. I actually feel that if the authors could restrict the patient population to include patients from 40+ this would make more clinical sense to me in regards to the purpose - probably this will only discard 2-4 patient in total and will only strength the conclusion.
Response: Thank you for this cause for thought. The median age of 58 years is also in line with other clinical studies regarding primary cementless THA of our institution. Our collective includes 5 patients < 40 years. In the view of the authors, adjusting the number of cases would limit the strength of the study. We hope Reviewer 1 agrees with this suggestion.
4.) Reviewer 1: Why is there no p-values in the last 2 parameters in table 1?
Also please remove redundant info in table and text.
Response: Thank you very much for the important hint. P-values of table 1 were added and redundant data were removed.
5.) Reviewer 1: I need the authors to comment in the discussion on whether the PROMs used actually captures what hip abductor deficiency leads to in elderly patients.
Response: Thank you for this valuable objection. To address this aspect, we have added the following section (Line 319-328: This does not reflect the results of the muscle force measurements. Both the HHS and the WOMAC score evaluate patient condition of daily living activities. The different increase in abductor muscle strength between the two groups may not be captured by the two scores and could possibly become apparent evaluating higher activity levels. However, pre- and postoperative HHS scores are comparable with those published in the literature and both scores are among the most frequently used outcome parameters in the literature [20,19,6].)
6.) Reviewer 1: And especially do the authors need to reflect on the surgical approach used and how this could influence the findings. They describe that they use a transgluteal approach and then they perform a study to see if the femur replacement affects gluteal performance (what about the surgical trauma).
Response: Thank you for another great comment which helped us further to improve the manuscript. Please see the subchapter “surgery and implant characteristics”. Here, we added a more detailed description of the approach used in this collective to clarify that it is not a conventional lateral approach according to Bauer with extensive incision of the gluteal muscles but a modified direct lateral approach with restricted gluteal incision (Line 117-122: The gluteus medius was incised along the fiber course to a maximum length of 3 cm to protect the inferior branch of the superior gluteal nerve. The anterior third of the gluteus medius was detached together with the underlying gluteus minimus ventrally to expose the joint capsule. Lenghtening of the incision into the vastus lateralis was strictly avoided.) Against the background of an approach-related muscular damage, there are studies in the literature indicating that a muscle sparing approach does not necessarily lead to a higher hip abductor muscle strength (Line 346-348: Furthermore, there are studies in the literature indicating that a muscle sparing approach does not necessarily lead to a higher hip abductor muscle strength one year postoperatively [21,22].). Nevertheless and according to the reviewer’s suggestion, this circumstance was discussed as a limitation (Line 335-348). We are aware that this approach does not conclusively correspond to a modern short stem concept.

Reviewer 3 Report
Thank you very much for your effort to improve the manuscript. However, I am afraid the authors did not answer and change their manuscript as I expected. First, I pointed out that stem's bony ingrowth to femur in their cases must be evaluated but authors showed only the references not their results. Second, the authors mentioned this is a part of RCT study and they included only first 70 cases despite their RCT was planed as 140 cases would be needed. The authors also mentioned the sample size of this study is relatively small. if so, they should show their results after they complete this RCT.
Author Response
Point-by-point response to the comments of Reviewer 3 on the manuscript “Abductor muscle force after straight-stem compared to short-stem total hip arthroplasty: Functional assessment of 70 patients of a randomized controlled clinical trial”
(invited resubmission jcm-1091637 – ROUND 2)
Specific comments Reviewer 3:
Reviewer 3:
Thank you very much for your effort to improve the manuscript. However, I am afraid the authors did not answer and change their manuscript as I expected. First, I pointed out that stem's bony ingrowth to femur in their cases must be evaluated but authors showed only the references not their results. Second, the authors mentioned this is a part of RCT study and they included only first 70 cases despite their RCT was planed as 140 cases would be needed. The authors also mentioned the sample size of this study is relatively small. if so, they should show their results after they complete this RCT.
Response:
- Thank you very much for your comment. With regard to the osseointegration of the femoral component, we tried to point out that previous studies could show that the axial subsidence is comparable to conventional stems, illustrating a similar bony ingrowth (see response letter ROUND 1). Therefore, we initially did not further highlight this aspect. Nevertheless, we agree with Reviewer 3 that the information about sufficient bony ingrowth without consecutive subsidence is highly relevant in terms of abductor muscle tension. To assess the degree of bony ingrowth, patients received radiological examinations on a routine basis 5 days, 3 months and 12 months postoperatively. We added this information and changed the manuscript accordingly (Line 204-208: All patients underwent radiological follow-up examinations on a routine basis 5 days postoperatively as well as 3 and 12 months after THA. The radiological follow-up included standardized digital anteroposterior (AP) radiographs of the pelvis and a lateral view of the affected hip. All radiographs were examined for signs of loosening, for example, radiolucent line formation, respectively bony ingrowth; Line 273-274: Within the follow-up period, there were no signs of loosening or osteolysis on conventional X-ray-evaluations. Bony ingrowth without stem subsidence was observed in all cases.)
- We clearly see the Reviewer’s point, thank you for your comment. With regard to the study design, it has to be noticed that the primary question of the mentioned RCT was to investigate the changes in bone mineral density after implantation of a cementless short compared to a conventional stem. When planning the study, we assumed a greater difference in means of the two groups for muscle force than for the DEXA measurements. Therefore, the number of cases for the bone mineral density analyses differ. We solely assessed the first 70 patients of the above mentioned RCT in terms of abductor muscle force. As previously mentioned in our response in ROUND 1 and against this background, the following facts have to be taken into account: Study designs of other RCT`s dealing with similar questions are comparable to the present study in terms of patient numbers (Jensen C et al. “Recovery in mechanical muscle strength following resurfacing vs standard total hip arthroplasty – a randomized clinical trial”, Osteoarthritis Cartilage 2011, DOI: 10.1016/j.joca.2011.06.011; Krych A J et al. “No benefit of the two-incision THA over mini-posterior THA: a pilot study of strength and gait”, Clin Orthop Relat Res. 2010, DOI: 10.1007/s11999-010-1660-6; Winther S B et al. “A randomized controlled trial on maximal strength training in 60 patients undergoing total hip arthroplasty”, Acta Orthop. 2018, DOI: 10.1080/17453674.2018.1441362.) Taken together, we hope to meet the reviewer’s expectations by addressing the above mentioned issues and thank you once again for your time and effort which helped us to substantially improve our manuscript.
